# Veterinary Telemedicine in Lithuania: Analysis of the Current Market, Animal Owner Knowledge, and Success Factors for Digital Transformation of Clinics

**DOI:** 10.3390/ani14131912

**Published:** 2024-06-28

**Authors:** Dalia Juodžentė, Evelina Burbaitė, Rolandas Stankevičius, Birutė Karvelienė, Jūratė Rudejevienė, Asta Daunorienė

**Affiliations:** 1Dr. L. Kriaučeliūnas Small Animal Clinic, Faculty of Veterinary, Veterinary Academy, Lithuanian University of Health Sciences, 47181 Kaunas, Lithuania; 2Faculty of Economics and Business, Kaunas University of Technology, 44029 Kaunas, Lithuania; 3Neurology and Neurosurgery Division, San Marco Veterinary Clinic, 35030 Veggiano, Italy; 4Animal Nutrition Department, Faculty of Veterinary, Veterinary Academy, Lithuanian University of Health Sciences, 47181 Kaunas, Lithuania

**Keywords:** veterinary telemedicine, online consultation, digital transformation

## Abstract

**Simple Summary:**

Both human and veterinary medicine are progressively evolving and constantly upgrading the quality of services provided. In the modern age, multiple solutions are available for digital transformation and for providing successful online veterinary telemedicine services. To better assess the situation that veterinary medicine is currently in, research is needed to evaluate the market for online veterinary services and the opinions of pet owners and veterinary clinic managers. This study summarizes and establishes the success factors of digital transformation from the perspective of the client, clinic owner, and manager; assesses the problems clients face with remote and contact consultations; and discusses the influence of respondent age and sex towards opinion on veterinary telemedicine services in Lithuania.

**Abstract:**

Veterinary telemedicine is used to provide animal health care information, education, and care remotely. Digital transformation of veterinary clinics is a fundamental process for the evolution of telemedicine and is changing the way veterinary care and animal health services are delivered. This study aimed to evaluate the current televeterinary market and to assess the knowledge of animal owners and clinic managers. The goals of this study were met by conducting an analysis of the current televeterinary market through a pet owner (*n* = 200) survey as well as clinic owner (*n* = 5) interviews. In contrast to other countries, only 1.85% of Lithuanian veterinary clinics offered a paid veterinary remote consultation service on their websites. In addition, more than half of animal owners who participated in the survey did not even know that remote veterinary consultations existed. The most important established factors for the success of digital transformation of veterinary telemedicine were investments in the marketing of the services, management leadership, the competencies and experience of veterinarians, convenient working hours for the client, faster service availability, and lower price. It is recommended to involve marketing and information technology company professionals and to invest in the specialty, digital, and communication skills of veterinarians.

## 1. Introduction

Veterinary telemedicine and telehealth are tools that help modernize the field by delivering animal healthcare and veterinary services online. In particular, both the American Veterinary Medical Association (AVMA) and the Canadian Veterinary Medical Association (CVMA) published guidelines for the use of telehealth in veterinary practice. It is the source from which the terms used worldwide derive their definitions. Telehealth is an umbrella term used to describe technology use for patient care and treatment, which can be general or personal (if a veterinarian–client–patient relationship (VCPR) is already established). Telemedicine (also referred as televeterinary practice), on the other hand, is used to communicate and share patient information remotely [1,2,3]. Both processes remain to be fully integrated into daily practice, with one of the reasons being that veterinary clinics are prone to invest into the competencies of veterinarians rather than infrastructure development. Owners of veterinary clinics do not dare to invest in digital transformation because of the impending changes, which cause confusion and employee dissatisfaction. Veterinarians are used to face-to-face consultations, and employers are not eager to change their approach. Some veterinarians even lack knowledge about telemedicine and its use in veterinary medicine [4]. The uncertainties of whether the services would be attractive to the user and whether the investments in digitalization would pay off are also demotivating the clinic owners. Information is lacking on how to accomplish successful digital transformation that would create added value, retain existing customers and clients, and attract new ones. It is also unclear whether public expectations for veterinary telemedicine align with those of veterinary business managers.

The studies on veterinary telemedicine are controversial. Some veterinary practitioners see it as a tool that makes it possible to visit a patient online, in a stress-free environment. Some studies also suggest that the patient’s recovery is easier to follow remotely [5]. Bragg et al. (2015) concluded that the use of telemedicine during scheduled consultations would help animals avoid the stress caused by traveling or by the veterinary clinic environment [6]. According to Bishop et al. (2021), both physicians and clients were satisfied with remote consultations. During their research, 72.6% of veterinarians spent equal or less time consulting remotely compared to contact consultation. About two-thirds had few or no problems connecting to the remote consultation [7]. The scientific literature reports that the advantages of telemedicine are better access to veterinary healthcare services, a closer connection between the veterinarian and the client, a lesser workload for the reception staff, and convenience for the client [8,9]. In Portugal, more than half of veterinarians agreed that teleconsultations can improve animal healthcare [10]. Despite that, another study suggested that up to 60% of remote veterinary consultation cases could end in a recommendation to visit a veterinary clinic [3].

Despite multiple benefits, online consultations can also have negative impacts on the patients. Healthcare issues might arise if animals are being diagnosed and treated medically without ever visiting them and establishing a VCPR. Different studies have expressed different opinions on whether the relationship must be formed prior to going online or if it can be formed remotely [1,2,3]. To date, studies assessing the VCPR are lacking, especially in Europe. Existing veterinary medicine standards could suffer, and both diagnostics and treatment protocols could be deviated from, leading to misdiagnosis. Diez et al. (2023) concluded that a third of owners were worried that they would not be able to assess their pets’ clinical condition at home during a remote consultation when asked by the veterinarian. It was also found that more than 82% of clients refused remote consultations due to limited treatment options [11]. Nonetheless, in a study conducted by Roca et al. (2019), up to 82.4% of all treatments prescribed for the animal during the remote consultation were correct [3]. However, because of increased symptomatic treatment, the excessive and irrational use of antibiotics could increase further [12,13].

Studies have concluded that differences between in-person and virtual communication is a further drawback of online consults, as are technology limitations [14,15]. AVMA guidelines state that connectivity is crucial for working remotely, and attention must be continuously paid to the quality and reliability of internet service and the availability of adequate bandwidth, resolution, and speed for clinical consultations [1].

The need for telemedicine in the veterinary field has grown strongly in Lithuania, as well as in the whole world. The special attention and need for digital services during the COVID-19 pandemic affected not only the medical but also the veterinary field. As technologies change, social structures, business models, knowledge, skills, professions, consumption culture, and lifestyles are transformed, and society’s expectations, especially for digitalized services, are increasing [16]. Telemedicine was rarely studied before the year 2019, but it all started with phone calls [17]. Since a uniform, multinational analytical study is difficult to perform, separate countries are conducting studies on a national level. In Portugal, 41 veterinarians were chosen for a survey, and they had a uniform opinion that restrictions to remote veterinary practice should be reduced, while guidance and regulation should be improved [7]. In Germany, 169 veterinarians participated in a survey that revealed how a majority of the doctors (77.5%) were insufficiently informed about current telemedicine opportunities. The lack of a clear legal framework was also an important factor in the scarce usage of remote teleconsults in Germany [18]. According to Mars and Auer (2006), telemedicine has been used in veterinary medicine for some time now, even though it is not yet sufficiently studied and the legal framework for veterinary telemedicine has not yet been settled or regulated in some countries, including Lithuania [12]. The purpose of this research is to further study consumer knowledge, worries and motivation factors regarding televeterinary services; the market; and veterinary clinic digitalization success factors. The goals were subdivided into categories:–To analyze the current market of veterinary telehealth providers in Lithuania;–To assess the animal owners’ knowledge on veterinary telehealth and establish motivational factors for its usage;–To assess clinic owners’ opinion on telehealth.

## 2. Materials and Methods

The study protocol was approved by the Ethics Committee of the Lithuanian University of Health Sciences (protocol code No. 2024-BEC3-T-014, approved on the 17th of May 2024). The study was conducted from April to May of 2024 and consisted of three parts. Questionnaires present in this study consisted of three types of questions: open ([open]), yes/no ([y/n]) or Likert scale ranging from 1 (strongly disagree) to 5 (strongly agree) ([Likert]). In light of the subjective nature of some criteria, the evaluations were performed by the same researcher.

### 2.1. Analysis of the Lithuanian Veterinary Telemedicine Market

To characterize the supply of digital veterinary services, an analysis of the websites of veterinary clinics that provide telemedicine services was carried out. First, a Lithuanian business database was searched for the keywords “veterinary clinic” [19]. After all veterinary clinics in Lithuania were selected, their websites were checked for remote services. After establishing a list of the clinics that use telemedicine, their services were evaluated. An analysis instrument consisting of 10 components was constructed for this part of our research (see Table 1). The main goals were to evaluate website usability, possibility to register for a consult, and the amount of provided information (price, specialist qualification, and service description). Companies that provided remote services only, without the ability to follow up with a contact consultation, were excluded.

### 2.2. A Survey of Lithuanian Small Animal Owners

The survey was formulated so that it would assess each owner’s knowledge on remote veterinary consultations, as well as the fear and/or motivation of small animal owners to use online veterinary services. Two hundred respondents were chosen. Criteria for inclusion were chosen as follows: older than 18 years, keeping any kind of pet, and not a clinic employee. The survey took place in the Veterinary Faculty of the Lithuanian University of Health Sciences, specifically the Dr. Leonas Kriaučeliūnas Small Animal Clinic, which provides a variety of telehealth services. Respondents were asked to complete the questionnaire while waiting in the clinic waiting room for their pets’ visits. Each respondent was given the questionnaire in person with an explanation of the anonymity of the questionnaire. All the small animal owners who participated in the survey were informed about the conduct, objectives, and the purpose of the research. Since the goal was to assess the general knowledge, the owners did not receive any information regarding the subject prior to the survey. The completed questionnaires were placed in a closed survey collection box. The results of the questionnaires began to be processed when the respondents filled out the entire expected number of questionnaires (*n* = 200). The survey consisted of questions on 6 topics (A–F) (see Table 2 for details):

### 2.3. Survey of Veterinary Clinic Owners

Based on the results of the animal owner questionnaire, an expert survey of veterinary clinic owners was conducted. The clinics providing telemedicine services in veterinary medicine were chosen (see Section 2.1). The survey aimed to find out the opinions of clinic owners on success factors that lead to the successful digital transformation of veterinary medicine. To better indicate the clinics, a list was made and numbered alphabetically (A–Z). The survey consisted of two parts (A–B): information about the clinic and its online services, as well as success factors for online consultations. See Table 3 for the detailed questionnaire scheme.

### 2.4. Statistical Analysis

The customer survey data were recorded in Microsoft Office Excel 2019 (Microsoft Corporation, Washington, DC, USA). Quantitative values following a normal distribution are presented as the mean ± standard deviation (SD). Quantitative indicators with a non-normal distribution are presented as the median with the smallest and largest values. Differences in opinions expressed for statements in parts C, D, and E of the customer questionnaire were assessed by ANOVA with Fisher’s LSD criterion. The distributions of the answers to the questions in part B and the differences of opinion were evaluated by the chi-square (χ^2^) test. Relationships between respondents’ social characteristics and expressed opinions were evaluated with the chi-square (χ^2^) test. Interview answers were processed using the “MAXQDA” program. Statistical data analysis was performed with the IBM SPSS Statistics^®^ software package (International Business Machines Corporation, New York, NY, USA), version 29. Statistically significant differences were considered to exist when *p* < 0.05.

## 3. Results

### 3.1. Analysis of Lithuanian Veterinary Telemedicine Market

By entering the keywords “veterinary clinic” into the Lithuanian business database, 378 companies engaged in veterinary activities were found. Of all the clinics found, a total of 7 (1.85%) internet pages indicated information about the veterinary telemedicine services provided. It was found that out of seven companies, five were small veterinary clinics that also provided contact veterinary consultations. The remaining two were online-only platforms/companies providing remote veterinary services.

A. Website design and use.

It was found that 71.4% (*n* = 5/7) websites were useful, eye-catching, and easily usable. They provided information on pet disease, treatment, and prevention; the website design and marketing solutions were adapted and grabbed the customer’s attention. A majority (85.7%, *n* = 6/7) of the websites were easy to understand and user-friendly.

B. Remote consultations on the website.

It was found that only 28.6% (*n* = 2/7) veterinary clinics websites met all three evaluation criteria (clear website menu, easily understandable reservation system, presented contacts of the veterinarian). The remaining 42.9% (*n* = 3/7) clinics did not meet any of the criteria.

C. Remote consultation online reservation system.

An online reservation system for remote consultations had not yet been implemented in 42.9% (*n* = 3/7) of clinics. For two (28.6%) veterinary clinics, an online reservation system was partially implemented. For the other two clinics (28.6%), the reservation system was easy to use, the menu was clear and additional information was provided to make the process user friendly.

D. Remote consultation implementation platform.

Two (28.6%) veterinary clinics did not indicate which platform was used for the remote consultations. Three veterinary clinic websites (42.9%) indicated that teleconsultation would be conducted via video call, one company (14.3%), conducted teleconsultation via e-mail, and one (14.3%) conducted it via phone call.

E. Prices and payment information.

It was noticed that 57.1% (*n* = 4/7) companies indicated neither the price of the consultation nor the payment options; 14.3% (*n* = 1/7) did not specify the price of the remote consultation; 28.6% (*n* = 2/7) did not indicate the payment options. The prices of remote veterinary consultations ranged from 19 to 150 euros, depending on the competence of the doctor and the city where such consultation was provided.

F. Information on veterinarians.

More than half (57.1%, *n* = 4/7) of the web pages did not clearly provide information about the competencies of veterinary doctors—specialists.

G. Review section and feedback on remote consultations.

When evaluating the feedback on remote consultations, it was noticed that 57.1% (*n* = 4/7) websites did not have a review section, and 42.9% (*n* = 3/7) had no visible reviews from other customers.

H. Remote services.

Only 28.6% (*n* = 2/7) of websites had information that displayed service descriptions. Other web pages (71.4%, *n* = 5/7) provided a general menu of services provided by the clinic.

I. Provision of remote services in English.

Only 28.6% (*n* = 2/7) of veterinary clinics’ web pages provided remote consultation information in English and the descriptions were identical to the Lithuanian remote consultation descriptions.

After evaluating all the web pages of veterinary clinics and their adaptation for remote consultations, it was found that only five clinics in Lithuania provided veterinary telemedicine services. Of them only one company has adapted the online platform for remote consultations to the client’s needs.

### 3.2. A Survey of Lithuanian Small Animal Owners

During the research, two hundred survey questionnaires were prepared, of which one hundred seventy were completely filled out, twenty were partially filled out, six were damaged due to incorrect marking, and four were refused by the owners.

A.Socio-demographic information.

Most respondents were women (72.4%). The mean age of respondents in this study was 37.9 ± 12.4 years. Respondents’ age was divided into three groups: 61.7% of the respondents belonged to the 18–40 age group, 34.2% of the respondents were 41–60 years old, and 4.1% were 61 years old and older. Slightly under two thirds (62.9%) of respondents had higher education. A majority of the respondents (80%) were working or were working and studying. Slightly over half (58.2%) of respondents indicated that they had used human medicine remote consultation services before. Around one fifth (21.7%) indicated that they had not used telemedicine services in human medicine but would like to do so. The remaining 20.1% of the respondents had not used remote consultation services in human medicine and did not plan to do so.

B.Information about owned pet(s).

Regarding the number of pets, 62.3% of respondents had one, while 37.7% had multiple animals. In total 170 respondents had 281 animals. Slightly more pets (51.9%) were mixed-breed than purebred (49.1%). More than half of the respondents indicated that their animal currently experiences minor (31.7%) or significant health issues (30.6%), while 37.7% of respondents stated that their animals are currently healthy. The survey concluded that 44.7% of respondents needed assistance for their pet, that could have been provided during a remote veterinary consultation, while 55.3% did not need such help.

C.Respondents’ knowledge about veterinary telemedicine in Lithuania.

Based on the answers to the questionnaire, it was found that 55.6% of respondents (*n* = 95) did not know that remote veterinary consultations even exist. Moreover, 57.1% did not know about the existence of remote veterinary consultations in Lithuania. Slightly over half (51.2%) of respondents were not able to mention a single clinic providing remote veterinary consultations, 25.3% knew at least one such clinic, and the remaining 23.5% could name two or more veterinary clinics providing remote consultations. Out of 170 respondents, 13.5% have used remote consultations offered by veterinary clinics. Televeterinary service knowledge was statistically more common in 18–40-year-olds than in other age groups (*p* < 0.001).

Twenty-nine (17%) respondents knew about remote consultations from their veterinarians. Websites of veterinary clinics were the source of remote consultation advertising for 20% of respondents. Twenty-two (12.9%) respondents learned about this service from social networks, and 10.6% heard about it through their colleagues, friends, or neighbors.

Thirty-six (21.2%) respondents believed that remote consultation had the same value as contact consultation, 62.9% believed that remote consultation was less valuable, and 15.9% respondents did not have an opinion on this issue. Most of the respondents (70%) believed in the usefulness of remote veterinary consultations, 10.6% thought that this type of consultation was not useful, and 19.4% had no opinion on the matter. During the investigation, it was found that 79.7% of respondents with one dog found remote veterinary consultations useful.

The Microsoft Teams app was preferred in our study, with 24.9%, followed by Facebook with 21.3% and Zoom with 20.9%. WhatsApp (12%), Google Meet (9.6%), and Skype (4.8%) was perceived as less popular and convenient. Only 5.7% did not have an opinion on online platforms for remote consultations.

We hypothesized that people who used human telemedicine services would be more likely to use remote veterinary services. Most respondents who used remote human medical consultations did not use remote veterinary consultations (*p* < 0.05).

D.Fears when choosing a contact veterinary consultation.

More than 32% of the respondents indicated that the waiting time at the veterinary clinic was a factor that made them unwilling to use contact consultations. The statement “Additional animal stress caused in the veterinary clinic” had significantly higher scores in female respondents compared to the males (*p* < 0.05). No significant differences between men and women were found when comparing other statements.

Comparing the research results between different age groups (18–40 years, 41–60 years, 61), statistically significant differences were observed in response to statements 3 (aggressive pet), 4 (distance to the veterinary clinic), and 5 (cost of the visit). The choice score in response to the statement: “Aggressive pet” was significantly higher in oldest age group compared to the middle-aged respondents (*p* < 0.05). Respondents aged 60 years and older would be afraid if they needed to take an aggressive pet to a contact veterinary consultation. They would probably prefer a remote consultation to avoid additional stress. Distance to the veterinary clinic and the time spent on the road, was perceived differently in all age groups. Younger respondents were significantly less demotivated if the clinic with the contact consultation was farther away, when comparing to the middle-aged respondents (*p* < 0.05). The oldest group of respondents was the most afraid of a long distance to the contact veterinary consultation and the time spent on the road. A significant difference (*p* < 0.05) was found between the oldest and youngest respondents. Scores of the statement “Cost of visit” were significantly different between youngest and oldest respondents. Older respondents were more afraid of the prices of contact veterinarian consultations compared to younger respondents (*p* < 0.05).

People who had used medical doctor online consultations before found contact veterinary visits more fearful. People who had never used human telemedicine and did not wish to try the services were less fearful. To them waiting time, animal stress, distance to the clinic, and visit price seemed less frightening in comparison to respondents who were users of human telemedicine services. The difference was statistically significant (*p* < 0.05).

E.Fears when choosing a remote veterinary consultation.

Respondents were partly intimidated by remote consultations, because during them the veterinarian is not able to examine the pet personally, and the respondent must provide the doctor with the necessary information. Animal owners were also partly intimidated by the fact that an insufficient treatment might be prescribed for the pet. Technical problems and inadequate internet connection or data protection issues would not have intimidated the service users.

Respondents of the youngest age group were statistically significantly more frightened of the veterinarian not being able to examine the pet personally, when comparing to the middle-aged respondents (*p* < 0.05). Inappropriate Internet connection during the consultation intimidated older survey participants statistically significantly more when comparing with the youngest and middle-aged respondents (*p* < 0.05).

F.Motivation to choose remote veterinary consultation.

Respondents would be partly motivated by the ability to receive a consultation from the desired specialist faster, to obtain a second opinion after sending medical history and exams, to have the consultation after working hours, and to avoid additional stress to the pet. Travel time saving and lower consultation prices were not essential factors influencing people to choose remote veterinary consultation. Lower consultation prices would motivate older age group respondents statistically significantly more compared to younger respondents (*p* < 0.05).

### 3.3. Survey of Veterinary Clinic Owners

Five clinics in total were selected because they provide remote veterinary services as well as contact consultations. One clinic owner refused to participate in the survey. The remaining 4 clinics were numbered A–D.

Veterinary clinic A. This clinic started providing veterinary services in 1994. In total, 19 veterinarians work in this clinic, of which 2 (specialists in veterinary neurology and dentistry) provide remote consultations. An online platform has been created for remote consultations, where the registration is available. It is possible to receive a remote consultation after working hours. Veterinarians provide remote consultations on the Vetforcall platform. After online registration, the customer is immediately directed to the payment platform. The head of the clinic noticed that the veterinary clinic conducts free remote consultations more often compared to paid consultations. Clients often bother doctors during non-working hours through personal social accounts.

Veterinary clinic B. This clinic started providing veterinary services in 2004. Started providing remote consultations in 2020. Only one veterinarian works in the clinic and provides remote consultations. The clinic has the option of registering for a remote consultation via the Internet, and the client chooses the time. Remote consultations can take place even after working hours, or at night. The veterinary clinic provides consultations on various platforms: by phone, e-mail, Facebook Messenger, WhatsApp, Viber, and Zoom. The clinic also provides free remote consultations. Clients often bother the doctor during non-working hours through personal social accounts.

Veterinary clinic C. This clinic started providing veterinary services in 1995. The clinic employs 32 veterinarians. Remote consultations are provided by different specialists: neurologists, internal medicine doctors, orthopedists, dermatologists, gastroenterologists, soft tissue surgeons, and specialists in minimally invasive surgery. The remote consultation service was launched in 2021 using the BBB online platform. Online registration for the consultation is not available. Remote consultations are available from 8:00 a.m. to 8:00 p.m. daily. At night, an on-call doctor gives free phone consultations, and there is an opportunity to come in for a contact veterinary consultation. The veterinary clinic conducts free remote veterinary consultations often because clients contact the veterinarians personally through their social accounts. Feeling constant pressure, the doctor is forced to respond to clients’ requests.

Veterinary clinic D. This clinic started providing veterinary services in 2013. Five veterinarians work in the clinic, of which 1 provides remote consultations. Started providing remote consultations a month ago. It is possible to register for online veterinary consultations, as well as choose the time of the visit. Clients do not have access to remote consultations after working hours. The veterinary clinic provides remote consultations by phone or on the Zoom video call platform. The head of the clinic stated that paid remote consultations are rarely carried out, as the clinic has only recently started providing these services. Veterinarians rarely give free remote consultations, lest clients disturb doctors through social accounts after working hours.

Time, speed, and cost of remote veterinary consultations were identified as three success factors that influence successful digital transformation in veterinary telemedicine. From the client’s perspective, employee competencies and experience were identified as two most important success factors for veterinary telemedicine improvement and expansion. From perspective of organizational strategy, marketing, management leadership, and active organization of processes were decided to be three most important success factors for veterinary telemedicine successful digital transformation. Three of the four companies stated that they evaluate the added value created by remote consultations according to the following indicators: the number of remote consultations, the number of contact consultations (if the client registered for it after the remote consultation), optimization of human resources, and the number of customer visits to the clinic’s website. The managers of the clinics noticed that a veterinarian only is needed for remote consultations, therefore the working time of the service staff is saved. In addition, it is thought that a remote consultation brings more profit to the clinic compared to a contact consultation because the clinic’s resources are saved.

## 4. Discussion

Telemedicine is slowly starting to change and challenge veterinary service providing businesses, but it seems that veterinary telehealth is still not implemented well enough in a majority of the countries. In a study by Dietz et al. (2023), slightly over 11% of those interviewed had already used remote veterinary services in Germany [11]. In another study it was determined that only 12% of dog breeders and 6% of cat breeders used telemedicine services in Austria, Denmark, and United Kingdom altogether [20]. Even though major studies are needed to assess the utility, usefulness, and application of telemedicine in veterinary field, this research provides a few novel insights on Lithuanian televeterinary. Only 1.85% of veterinary clinics here provide remote veterinary services. In addition, 57.1% of respondents in our study did not know about the existence of remote veterinary consultations in Lithuania. Only 13.5% of respondents in our study have used remote consultations offered by veterinary clinics before, while 70% believed in the usefulness of televeterinary.

A study by Diez et al. (2023) has raised a hypothesis that animal owners are beginning to be disappointed with contact consultations at present. Two-thirds of customers were dissatisfied with the waiting time at the veterinary clinic, and about 30% were dissatisfied with the distance to the veterinary clinic and the time spent on the road. More than half of respondents who had an aggressive pet were unable to control it, and the visit was stressful to the pet, the owner, and the doctor [11]. Understanding the different needs of the animals, many exotic animal practitioners as well believe that remote consultation would be useful when providing the client with information about the exotic pet‘s housing conditions, feeding, and care [21]. A study by Bishop et al. 2018 aimed to compare contact and remote post-operatory control visits. Dog owners in both groups were satisfied with the visits, but those in the telemedicine group found their pets less fearful and stressed; therefore, they planned to continue using remote veterinary services [22].

The most common fears that animal owners experienced during a teleconsult, were the fear of an inaccurate diagnosis and the fear of insufficient treatment. This could explain why in 60% of cases, remote veterinary consultations end up with a recommendation to visit a doctor in person [8]. However, scientific literature suggests that even 82.4% of the treatments prescribed for the animals during the remote consultation were correct in the United States of America (USA) [3].In the current study, only 10.6% of respondents thought that remote consultations are not useful.

The prices of remote veterinary consultations in Lithuania ranged from 19 to 150 euros, depending on the competence of the doctor and the city where the consultation is provided. Some consultations were given for free, and the teleconsults that cost 150 euros were priced this high to allow the most urgent cases to be consulted in a timely manner. Veterinary research states that 20% of veterinarians doubted whether clients would be willing to pay for remote veterinary services [23]. Another study concluded that an owner would be willing to pay about 40 pounds (46.77 euro) for a remote veterinary consultation [24]. Widmar et al. (2020) found that American dog owners would be willing to pay $38.04 more for a remote veterinary consultation with an outside specialist and $13.38 more for a consultation with their own veterinarian, compared to regular face-to-face veterinarian consultation prices. Cat owners also indicated that they would be willing to pay more for remote visits of both an outside specialist veterinarian and a local veterinarian [25]. Recent research states that the cost of remote consultation was the least problem that would make veterinary clinic clients refuse the service [11].

Some significant differences in provided answers were found between different age groups. In our study, the oldest group of respondents was the most afraid of the consultation price, a long distance to the contact veterinary consultation, and the time spent on the road. Recent study states that a fourth of clients would not have been able to secure transportation for an in-person veterinary clinic consult [26]. It might be explained by the fact that visits at the clinic require more time and financial resources.

Televeterinary service knowledge was better between 18–40-year-olds in comparison to other age groups. Scientific literature suggests that respondents aged 30–44 knew the most about remote veterinary consultations [25]. This age group division is different than the one we used, but 30–44 age group falls in our category of 18–40-year-olds. This scientific data is in agreement to our study. It is also reported that the younger (18–29) and middle-aged (30–44) respondents had a significantly positive attitude towards televeterinary, compared to the older (45–59 and 60–75) group respondents [25]. Online services are usually more popular within young citizens; therefore, the current statistics are in agreement with the demographic data as well.

People who have never used human telemedicine were less worried about the waiting time, animal stress, distance to the clinic, and visit price of a contact visit at the clinic in comparison to respondents who were users of human telemedicine services. During the pandemic of COVID-19, the healthcare of many countries was challenged and later improved. For example, in Argentina, human telemedicine was mostly used my young, healthy patients in their thirties, until the necessary changes were made [27]. Inappropriate Internet connection during remote veterinary consultation intimidated the older survey participants more than others. In addition, studies conclude that the use of healthcare information technologies was significantly lower among the age groups 65 or older compared with the younger age groups [28]. We can assume that older patients have a harder time coping with modern technology. Studies indicate that only a tenth of the respondents were worried about technical problems and insufficient internet connection during the consultation. Participants of all age groups were not afraid of inadequate data protection during remote consultations. Properly ensured data protection was the least important problem seen by clients preparing to use a remote veterinary consultation service [11].

Overall, despite the scarce usage and lack of legislation of telehealth in Europe and USA, it seems to bring benefits to the veterinary world [1,2,11,20]. The usage of teleconsults is growing and results in good outcomes. Recent study concludes that clients who had a remote consultation for veterinary rehabilitation were as satisfied as the ones who underwent an in-person visit. The teleconsult group were even happier with the simplicity of the procedure [29]. Multiple studies evaluated the promising veterinary healthcare changes during the COVID-19 pandemic [7,27,30,31]. It was concluded that as much as 89% of clients that used televeterinary visits were either satisfied or very satisfied with the care that their pets received [30]. Therefore, we believe that televeterinary use is growing and will continue to grow, as long as the specialists in the field are willing to go online. To ensure the successful integration of remote veterinary services in clinics, the following can be recommended:(1)Integrate remote veterinary services by involving external partners (marketing specialists, IT companies) who would ensure timely help and advice in creating a remote consultation platform and in creating advertising strategies;(2)Invest in the specialty, digital, and communication competencies of the veterinarians in the company, so that the telemedicine services would be attractive services for consumers;(3)Invest in equipment (computer, video, and audio equipment) that is necessary to ensure quality telemedicine services;(4)Equip a professional-looking room where telemedicine services will be provided.

We acknowledge, that the current study has several limitations. The study was conducted with Lithuanian veterinary clinics only, so a similar study could be conducted in other countries by interviewing the pet owners and veterinary clinic managers and assessing the televeterinary market. What is more, it is a first study evaluating Lithuanian veterinarian telehealth. Quite unexpectedly, the market analysis resulted in a very small sample size (five clinics), consisting entirely of small animal clinics. In addition, the survey of animal owners was performed in one clinic only. It is likely that the statistical data do not represent the veterinary market and the average Lithuanian animal owner. To confirm our findings, additional studies might be needed.

Several of the selected analysis criteria within the study are subjective and open to interpretation; therefore, the same researcher was chosen to evaluate the webpages and interviews. We cannot exclude that the results and data interpretation were somewhat influenced by personal views.

## 5. Conclusions

Veterinary telemedicine in Lithuania is still rarely used; animal owners do not have enough knowledge about it, and clinic managers seldomly decide to invest in remote veterinary services. However, respondents did appreciate telehealth benefits such as reducing animal stress, saving time spent on the road and at the clinic, and the possibility of receiving specialized consultation faster and at a comfortable time. Managers of clinics that provided veterinary telemedicine services stated that success factors for smooth digitalization of the clinics included the time, speed, and cost of remote veterinary consultations. Doctor competencies and experience were acknowledged as the most important factors for the clients.

## Figures and Tables

**Table 1 animals-14-01912-t001:** Web page analysis instrument. All of the questions were evaluated as yes/no [y/n].

Analysis Components	Questions
A. Website design and use	A.1 Useful
	A.2 Comfortable
	A.3 Easily understandable
	A.4 Eye-catching
B. Remote consultations on the website	B.1 Clear menu
	B.2 Easy reservation system
	B.3 Information on how to contact a veterinarian
C. Remote consultation online reservation system	C.1 Direct remote consultation reservation through the website
	C.2 Ability to choose time and specialist
	C.3 Ability to provide information about a pet
D. Remote consultation implementation platform	D.1 Remote consultation is carried out by video call
	D.2 Remote consultation is carried out by e-mail
	D.3 Remote consultation is carried out by phone call
	D.4 Remote consultation is carried out by social networks
E. Prices and payment information	E.1 The price of the consultation is clearly indicated
	E.2 Clearly presented information about payment execution
F. Information on veterinarians	F.1 Clearly presented information on the competence of veterinary specialists
G. Review section and feedback on remote consultations	G.1 A clearly indicated place for leaving a review
	G.2 Reviews of other customers are visible
H. Remote services	H.1 A clearly presented menu of remote services
	H.2 Description of services provided
I. Provision of remote services in English	I.1 Information about performed remote services is also provided in English
	I.2 The English and Lithuanian service descriptions are identical

**Table 2 animals-14-01912-t002:** Animal owner survey scheme. The scale for Likert questions was 1–5 and ranged from ”strongly disagree” to “strongly agree”.

Analysis Components	Questions
A. Socio-Demographic Information	1. Sex [open]
	2. Age [open]
	3. City of residence [open]
	4. Social status [open]
	5. Education [open]
	6. Have you used remote consultations in human medicine? [y/n]
B. Information About Owned Pet(s)	1. How many animals do you raise? [open]
	2. What breed of dog/cat (other species) do you keep? [open]
	3. How old is your pet? [open]
	4. What is your pet‘s sex? [open]
	5. Does your pet currently have health problems? [y/n]
	6. Has your pet required veterinary assistance that could have been provided by a veterinarian during a remote consultation? [y/n]
C. Respondents’ knowledge about veterinary telemedicine in Lithuania	1. Did you know that you can consult with a veterinarian during a remote consultation? [y/n]
	2. Until now, have you heard/knew that you can consult a veterinarian remotely in Lithuania? [y/n]
	3. How many veterinary clinics do you know that offer remote consultations to clients? [open]
	4. Have you used the remote consultation service of veterinary clinics? [y/n]
	5. Has your veterinarian previously mentioned to you that you can use the remote consultation service? [y/n]
	6. Do you think that a remote consultation with a veterinarian is of the same value as a face-to-face consultation? [y/n]
	7. Do you think that remote consults in veterinary medicine are useful? [y/n]
	8. Which online platform would be the most convenient for you if you were to use the remote veterinary consultation service? [open]
	9. How did you find out about the possibility of remote veterinary consultations? [open]
D. Fears when choosing a contact veterinary consultation	1. Waiting time at the veterinary clinic [Likert]
	2. Additional stress in the veterinary clinic [Likert]
	3. Aggressive pet [Likert]
	4. Distance to the veterinary clinic and time spent on the road [Likert]
	5. Cost of the visit [Likert]
E. Fears when choosing a remote veterinary consultation	1. The veterinarian will not be able to examine the pet with his own hands [Likert]
	2. I will not be able to provide the doctor with the necessary information about my pet with the means I have at home [Likert]
	3. Inadequate treatment will be provided to the pet [Likert]
	4. Technical problems during the consultation [Likert]
	5. Bad internet connection [Likert]
	6. Concerns about inadequate data protection [Likert]
F. Motivation to choose remote veterinary consultation	1. The possibility to get the consultation of the desired specialist faster [Likert]
	2. Possibility to save travel time [Likert]
	3. The pet will not be stressed [Likert]
	4. Lower consultation prices [Likert]
	5. The possibility of getting a second opinion after sending medical history and exams [Likert]
	6. Availability of after-hours consultation [Likert]

**Table 3 animals-14-01912-t003:** Veterinary clinic owner expert survey scheme. All questions were open-ended questions [open].

Part A	Part B
1. How many veterinarians work in your clinic?	1. Do you think free phone consultations would help to create added value to your clinic?
2. Is it possible to register for veterinary services online at your clinic?	2. How do you think the increased number of remote consultations could change the structure of the organization?
3. Is it possible to book an appointment online at your clinics?	3. Do you think that repeated remote consultations could optimize the clinic’s human resources?
4. How many veterinarians provide remote veterinary consultations in your clinic?	4. What added value does remote consultation in your clinic create for the veterinary services offered?
5. Does your clinic have the possibility to receive a remote consultation of veterinarians after working hours?	5. How do you measure the success of your company’s digital transformation and what indicators do you use to measure it?
6. Do you think that remote consultations before and after traditional working hours would be convenient for the client?	6. What do you think is the most important factor in the success of digital transformation in the context of remote veterinary consultations?
7. Does your clinic have the possibility to receive remote consultation of veterinarians on weekends?	7. What impact do digitized services have on your business strategy?
8. Do you think that the platform you use is convenient for the customer?	8. What influence does the competences of employees have on the success of digitization?
9. In your clinic, do you value remote consultation the same as contact consultation?	9. What influence does the organization of company processes have on the success of the company’s digitization?
10. How often do your veterinarians consult for free using various means of communication?	
11. How often do remote consultations take place in your clinics?	
12. Would getting a specialist consultation earlier than usual, affect the popularity of remote consultations?	
13. Are the doctors of your clinic often interrupted by customers during their non-working hours through social networks or calls?	
14. Do you think that a cheaper online consultation would be more popular than a contact consultation?	
15. What changes should be made to carry out more remote consultations?	

## Data Availability

Data are available on request due to privacy restrictions.

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
