# Peer review of "Veterinary Telemedicine in Lithuania: Analysis of the Current Market, Animal Owner Knowledge, and Success Factors for Digital Transformation of Clinics"

_animals, 2024, doi:10.3390/ani14131912_

Round 1
Reviewer 1 Report
Comments and Suggestions for Authors
The paper describes the problem of remote veterinary consultations in veterinary medicine which is not well recognized and described.
It's a valuable contribution to the discipline ov veterinary practice management and should be better studied in the future mainly due to the unsolved problem of payment for remote contact with veterinarian and consultations.
Author Response
The paper describes the problem of remote veterinary consultations in veterinary medicine which is not well recognized and described. It's a valuable contribution to the discipline ov veterinary practice management and should be better studied in the future mainly due to the unsolved problem of payment for remote contact with veterinarian and consultations.
We appreciate your valuable comments and opinion, we would like to thank you for your time and important contribution to improving our manuscript.
Reviewer 2 Report
Comments and Suggestions for Authors
See the attached file

Minor English editing is required
Author Response
We appreciate your valuable comments and opinion, we would like to thank you for your time and important contribution to improving our manuscript.
Here are the answers to the comments that you provided:
-Title:"The" and "And" should be "the" and "and": Thank you for the correction, we have made the change.
-Abstract lacks the methodology of the study: The abstract has been corrected and information was added. The changes have been highlighted in the edited manuscript.
-The status of veterinary telemedicine use across the globe is not described. We are thankful for your insights. We have revised the introduction and added information on international televeterinary studies.
-Line 61: Replace 66.2% by "About two thirds": We are thankful for your correction, the change has been made in the manuscript.
-Line 71: add more information about the importance of "veterinarianclient-patient relationship" and its status in Lithuania. For this purpose you can read this article: doi.org/10.5455/OVJ.2024.v14.i3.20: Thank you for the interesting comment. We think that being attentive to detail is very important, therefore revision and changes have been made to more carefully adress the veterinarian client patient relationship in the study. We have implemented the recommended study in the manuscript. However, it is a review paper, and does not provide novel information but rather discusses previously published data.
-Follow the format of the journal in writing of titles and subtitles: we are once again thankful for your insights. The changes have been made accordingly and highlighted throughout the manuscript.
-Line 117: Replace "200" with "Two hundred": the change has been made, thank you.
-In tables 1-3, please add the type of questions (open, yes/no or likert scale) in between brackets beside each question: thank you for your attention to detaills, the question type has been specified.
-Delete lines 145-148: the changes have been made, thank you.
-Line 149: Add a subtitle "Statistical analysis": The change has been made and highlighted.
-Line 155: ( 2)??: We intended the chi square test and a special character has been used in the manuscript that may not be supported in different programs. The change has been made.
-Line 174-176 (B. The availability……): these are not results. Please transfer this information to M&M section: Thank you, the phrase has been evaluated and transfered.
-Lines 222, 227, 230, 231, 237, 243, 335.. etc Please do not start the sentences with numbers: we are once again thankful for your insights. The changes have been made accordingly and highlighted throughout the manuscript.
-Lines 265-266: Please mention the statements (3-5): Thank you for the correction, we have made the change.
-This section is very weak and many sentences are results rather than discussion. You can improve Discussion by reading the recently published articles (2022-2024) in veterinary telemedicine: we are once again thankful for your insights. The changes have been made accordingly and highlighted throughout the manuscript.
-Line 360: Revise this sentence: Thank you for the correction, we have made the change.
-Lines 365-366: Revise this sentence "this research provides a few novel insights". At least, add "in Lithuania": We have corrected this sentence and specified the location of the study.
-Lines 366-367: Our results state…. These are results not a discussion. Please delete this sentence.: we are once again thankful for your insights. The results were moved to the results section and the discussion has been made more focused.
-Lines 371-371: Please add the countries of these studies ([9] and [16]): Thank you for the correction, we have made the changes.
-Few references are cited (n=21) and many recent references are missed. Please cite at least 30 references (9 more) concerning this hot topic, particularly recently ones (2020-2024): We agree with the need of additional references. The references have been added and corrected within the paper.
-Please follow the format of Animals in all references: We have changed the citation style within the manuscript. thank you for the valuable comment. We hope that after extensive revisions that we made, the manuscript will now be appropriate for acceptance for publication in the Animals journal.
We hope that after extensive revisions that we made, the manuscript will now be appropriate for acceptance for publication in the Animals journal.
We are thankfull for your valuable review.
Sincerely,
Evelina Burbaite.
Reviewer 3 Report
Comments and Suggestions for Authors
This research evaluates 1 consumer knowledge, worries, and motivation factors for televeterinary services, the 92 market and veterinary clinic digitalization success factors in Lituania country.
The manuscript is well organized and written, but a revision of language is required from a native speaker.
Comments on the Quality of English LanguageA revision of english language is required from a native speaker.
Author Response
The manuscript is well organized and written, but a revision of language is required from a native speaker.
We appreciate your valuable comments and opinion, we would like to thank you for your time and important contribution to improving our manuscript. The changes have been made accordingly and highlighted throughout the manuscript.